# Palbociclib Plus Fulvestrant or Everolimus Plus Exemestane for Pretreated Advanced Breast Cancer with Lobular Histotype in ER+/HER2− Patients: A Propensity Score-Matched Analysis of a Multicenter Retrospective Patient Series [note 1]

**DOI:** 10.3390/jpm10040291

**Published:** 2020-12-18

**Authors:** Armando Orlandi, Elena Iattoni, Laura Pizzuti, Agnese Fabbri, Andrea Botticelli, Carmela Di Dio, Antonella Palazzo, Giovanna Garufi, Giulia Indellicati, Daniele Alesini, Luisa Carbognin, Ida Paris, Angela Vaccaro, Luca Moscetti, Alessandra Fabi, Valentina Magri, Giuseppe Naso, Alessandra Cassano, Patrizia Vici, Diana Giannarelli, Gianluca Franceschini, Paolo Marchetti, Emilio Bria, Giampaolo Tortora

**Affiliations:** 1Comprehensive Cancer Center, UOC di Oncologia Medica, Fondazione Policlinico Universitario A. Gemelli IRCCS, 00168 Rome, Italy; elena.iattoni@unicatt.it (E.I.); carmela.didio@unicatt.it (C.D.D.); antonella.palazzo@policlinicogemelli.it (A.P.); giovanna.garufi@unicatt.it (G.G.); giulia.indellicati@unicatt.it (G.I.); alessandra.cassano@unicatt.it (A.C.); emilio.bria@unicatt.it (E.B.); giampaolo.tortora@unicatt.iT (G.T.); 2Division of Medical Oncology, Regina Elena National Cancer Institute IRCCS, 00128 Rome, Italy; pizzuti8@hotmail.com (L.P.); alessandra.fabi@virgilio.it (A.F.); patrizia.vici@ifo.gov.it (P.V.); 3Medical Oncology, Central Hospital of Belcolle, 01100 Viterbo, Italy; agnese.fabbri@yahoo.it (A.F.); danielealesini@yahoo.it (D.A.); 4Clinical and Molecular Medicine Department, Sapienza University of Rome, 00185 Rome, Italy; andreabotticelli@hotmail.it (A.B.); magri.v@hotmail.it (V.M.); Giuseppe.Naso@uniroma1.it (G.N.); paolo.marchetti@uniroma1.it (P.M.); 5Comprehensive Cancer Center Division of Gynecologic Oncology, Fondazione Policlinico Universitario A. Gemelli IRCCS, 00168 Rome, Italy; luisa.carbognin@policlinicogemelli.it (L.C.); ida.paris@policlinicogemelli.it (I.P.); 6Oncology Department, Ospedale di Frosinone, 03100 Frosinone, Italy; angelavaccaro64@gmail.com; 7Oncology Department, Azienda Ospedaliero-Universitaria Policlinico di Modena, 41125 Modena, Italy; l.moscetti@icloud.com; 8Medical Oncology, Department of Traslational Medicine and Surgery, Università Cattolica del Sacro Cuore, 00168 Rome, Italy; gianluca.franceschini@unicatt.it; 9Biostatistical Unit, Regina Elena National Cancer Institute IRCCS, 00128 Rome, Italy; diana.giannarelli@ifo.gov.it; 10Multidisciplinary Breast Center, Dipartimento Scienze della Salute della donna e del Bambino e di Sanità Pubblica, Fondazione Policlinico Universitario A. Gemelli IRCCS, 00168 Roma, Italy

**Keywords:** advanced breast cancer, mTOR inhibitor, CDK4/6 inhibitor, endocrine resistance

## Abstract

Cyclin-dependent kinase 4/6 inhibitors (CDK4/6i) in combination with endocrine therapy (ET) show meaningful efficacy and tolerability in patients with metastatic breast cancer (MBC), but the optimal sequence of ET has not been established. It is not clear if patients with lobular breast carcinomas (LBC) derive the same benefits when receiving second line CDK4/6i. This retrospective study compared the efficacy of palbociclib plus fulvestrant (PALBO–FUL) with everolimus plus exemestane (EVE–EXE) as second-line ET for hormone-resistant metastatic LBC. From 2013 to 2018, patients with metastatic LBC positivity for estrogen and/or progesterone receptors and HER2/neu negativity, who had relapsed during adjuvant hormonal therapy or first-line hormonal treatment, were enrolled from six centers in Italy in this retrospective study. A total of 74 out of 376 patients (48 treated with PALBO–FUL and 26 with EVE–EXE) with metastatic LBC were eligible for inclusion. Progression-free survival (PFS) was longer in patients receiving EVE–EXE compared with PALBO–FUL (6.1 vs. 4.5 months, univariate HR 0.58, 95% CI 0.35–0.96; *p* = 0.025). On the propensity score (PS) analysis, PFS was confirmed to be significantly longer for patients treated with EVE–EXE compared to PALBO–FUL (6.0 vs. 4.6 months, *p* = 0.04). This retrospective analysis suggests that EVE–EXE is more effective than PALBO–FUL for second line ET of metastatic LBC, allowing us to speculate on the optimal therapeutic sequence.

## 1. Introduction

Invasive lobular breast carcinomas (LBCs), which account for up to 15% of all invasive breast cancers (BC), are almost always estrogen-positive (ER, coded by the ESR1 gene) and lacking HER2 amplification and as such are treated with endocrine therapy (ET) [1]. Options for ET have expanded in the last two decades with the availability of new agents, including selective estrogen receptor modulators (SERM), aromatase inhibitors (AIs), and selective estrogen receptor degrader (SERD) [2,3]. However, resistance to therapy and subsequent disease progression continue to be major problems. More than a third of patients with ER-responsive early-stage BC and almost all of those with metastatic disease become refractory to these treatments during the course of their disease [4,5,6]. New approaches to treatment are clearly required, and to this end, cyclin-dependent kinase 4/6 inhibitors (CDK4/6i) were developed. CDK4/6i palbociclib, ribociclib, and abemaciclib in combination with ET have shown clinically meaningful efficacy and a good tolerability profile in patients with metastatic breast cancer (MBC), in endocrine sensitive and endocrine resistant disease, within the PALOMA, MONALEESA, and MONARCH trials, respectively [7,8,9,10,11,12]. The MONALEESA-3, MONALEESA-7, and MONARCH-2 trials showed significantly improved overall survival with a combination of a CDK4/6i and ET [10,13].

While subgroup analysis of the PALOMA 2 trial showed that the combination of palbociclib plus letrozole is effective in first-line treatment both in ductal and lobular histotypes, no evidence is currently available on the efficacy of CDK4/6i exclusively in second-line treatment according to histotype (PALOMA 3, MONARCH 2, and MONALEESA 3) [7,12,13]. Recently, a pooled analysis of seven phase III trials (combining the data of the endocrine sensitive and resistant setting) was made to investigate the benefit of adding CDKIs to endocrine therapy in patients whose tumors might have differing degrees of endocrine sensitivity, such as the lobular histotype [14]. This pooled analysis shows that all subsets, including LBC, of patients derived benefits from the addition of a CDKI to endocrine therapy.

For some time in our clinical practice, we have observed that patients with ER-positive metastatic LBC who had relapsed on adjuvant tamoxifen/AIs or had progressed with first-line hormonal therapy tended to show poor responses, and their disease showed faster progression with CDK4/6i [15]. Interestingly, in some of these patients, the subsequent use of a mTOR inhibitor (everolimus) produced greater clinical benefits and prolonged survival. In light of these considerations, we conducted a multicentric, retrospective study to compare the efficacy of the combination of palbociclib plus fulvestrant (PALBO–FUL) with everolimus plus exemestane (EVE–EXE) as second-line ET for hormone-resistant metastatic LBC.

## 2. Materials and Methods

This retrospective study enrolled women with metastatic LBC from six Italian oncology centers over a five-year period from 2013 to 2018. Female patients (≥18 years at diagnosis) with metastatic LBC (confirmed by metastasis biopsy) or with a clinical history of disease, compatible with recurrent lobular carcinoma of the previously diagnosed primary breast cancer, positivity for estrogen and/or progesterone receptors, and HER2/neu negativity, who had relapsed during adjuvant hormonal therapy or a first-line hormonal treatment, were eligible for inclusion. Patients were excluded if they relapsed in a period of more than 12 months from the end of adjuvant hormonal therapy or they had not received prior hormonal treatments. Patients received second line therapy with PALBO–FUL or EVE–EXE according to standard approved administration schedules. All patients enrolled in the study provided written informed consent for their data to be used for future medical research. The study was conducted in accordance with Italian legislation on observational studies (Min. Sal. Circular 6 September 2002). Data from the six participating centers were processed and stored at the coordinating center (Fondazione Policlinico Universitario Agostino Gemelli IRCCS, Rome, Italy) in compliance with local privacy regulations.

The primary endpoint was progression-free survival (PFS) defined as the interval between the treatment start date and the disease progression date. Secondary endpoints were objective response rate (ORR, rate of complete objective responses and partial objective responses of the disease to the treatment evaluated using clinical and/or radiological criteria, according to RECIST 1.1 Criteria) and clinical benefit rate (CBR, rate of complete objective responses, partial objective responses, and stable disease in response to the treatment evaluated with clinical and/or radiological criteria).

All continuous data were expressed as mean ±SD, range, and median value; frequencies and percentages were reported for categorical variables. The clinical, biological, and pathological characteristics of tumors at baseline were determined using Fisher’s exact test. PFS and overall survival were estimated by the Kaplan–Meier limit product method. The Cox regression model was applied to multivariate survival analysis, and *p* values and hazard ratios (HRs) with 95% CI were obtained. All significant variables in the univariate model were used to build the multivariate model of survival. A propensity score (PS) adjustment for baseline characteristics was conducted for survival analysis. The Statistical Package for the Social Sciences (SPSS) 20.0 software, (Chicago, IL, USA) was used for statistical analysis and integrated with Medcalc software V.9.4.2.0 (Mariakerke, Belgium). In all analyses, the significance level was specified as *p* < 0.05. As the study was explorative, an estimate of the sample size was not calculated.

## 3. Results

### 3.1. Patient Demographics

Of a total of 376 women screened over the five-year period (2013–2018) in the six centers, 74 were diagnosed with metastatic LBC. Of these, 48 patients received PALBO–FUL and 26 EVE–EXE. Baseline patient characteristics were comparable between the two treatment groups (Table 1). Most patients were post-menopausal (89% and 100% in the PALBO–FUL and EVE–EXE groups, respectively), had non-visceral disease (61 and 68%, respectively), and had less than three sites of metastasis (78 and 79%, respectively). Overall, 43 and 57% of patients in the PALBO–FUL and EVE–EXE groups, respectively, had previously received two lines of endocrine therapy, and 15 and 17% of patients, respectively, had metastatic disease on diagnosis. All patients had received at least one or two lines of endocrine therapy (aromatase inhibitors alone or in combination with tamoxifen, or fulvestrant).

### 3.2. Efficacy and Activity

Median PFS was significantly longer in patients receiving EVE–EXE than in those receiving PALBO–FUL (6.1 vs. 4.5 months, univariate HR 0.58, 95% CI 0.35–0.96; *p* = 0.025 (Figure 1)). Univariate analysis showed that metastatic stage at diagnosis (HR 2.82, 95% CI 1.43–5.56; *p* = 0.003), previous chemotherapy exposure (HR 0.41, 95% CI 0.24–0.72, *p* = 0.002), and study treatments (HR 0.58, 95% CI 0.35–0.96, *p* = 0.025), correlated positively with PFS (Table 2). On multivariate analysis, previous chemotherapy exposure was the only factor significantly associated with PFS (HR 0.41, 95% CI 0.24–0.72, *p* = 0.002).

PFS was significantly longer in patients receiving EVE–EXE in comparison with PALBO–FUL (6.0 vs. 4.6 months, *p* = 0.04) on PS analysis adjusted for prior chemotherapy and synchronous/metachronous metastatic status (Figure 2). Objective response rates in both groups did not significantly differ, with 7 out of 46 patients (ORR 15.2%, 95% CI 4.8–25.6) in the PALBO–FULV group and 9 out of 28 patients (ORR 32.1%, 95% CI 14.8–49.4) in the EVE–EXE group (*p* = 0.0725). Accordingly, no difference in CBR was found between both groups (PALBO–FULV 65.2%, 95% CI 51.4–78.9 and EVE–EXE 67.8%, 95% CI 50.5–85.1, *p* = 1.0) (Figure 3). Only 1 patient experienced a complete response (CR) in the PALBO–FULV group (CR 2%, 95% CI < 1–6.3). Stable disease (SD) and progressive disease (PD) were 35.7% (95% CI 17.9–53.4) and 32.1% (95% CI 14.8–49.4), respectively, in the EVE–EXE group, and 50.0% (95% CI 35.5–64.4) and 34.7% (95% CI 21.0–48.5) in the PALO–FULV group, respectively.

### 3.3. Safety/Adverse Events

In terms of safety and adverse events, both treatments were relatively well tolerated (Table 3). In the PALBO–FUL group, neutropenia (65%) and anemia (41%) were the most commonly reported events, while in the EVE–EXE group, fatigue (64%), stomatitis (35%), and rash (25%) were the most reported adverse events (Table 3). Grades 3 and 4 adverse events (in the main afebrile neutropenia) occurred in 24 patients (52%) in the PALBO–FUL group, and 6 patients (21%) receiving EVE–EXE reported grade 3 adverse events (stomatitis and cutaneous rash and one case of interstitial pneumonitis). Dose reduction was required in 11 (24%) of patients in PALBO–FUL and 12 (43%) in the EVE–EXE group. Treatment discontinuations were all subsequent to disease progression, except in one case—a patient who developed interstitial pneumonitis while receiving EVE–EXE discontinued treatment. No deaths related to study medications were reported.

## 4. Discussion

Despite the clinically meaningful efficacy and good tolerability profile of the combination of CDK4/6i and ET in patients with MBC, patients eventually experienced disease progression and the emergence of resistance [16]. Resistance to CDK4/6i plus ET represents the next clinical challenge for the breast cancer community to overcome and requires a deep understanding of the mechanism of CDK4/6i resistance in an endocrine sensitive and resistant setting. Furthermore, there are limited data on the efficacy of these treatments in different BC histological types, in particular in patients with LBC who are often not well represented in clinical trials. While a subgroup analysis of PALOMA 2 trial showed that the combination of palbociclib plus letrozole was effective as a first-line treatment both in ductal and lobular histotypes, and in BOLERO-2, everolimus was shown to be effective both in ductal and lobular histotype hormone-refractory patients [17], no evidence is currently available on the efficacy of CDK4/6i as a second-line treatment based on histotype [18]. Thus, the treatment options for this frequent BC subtype are limited if tumors develop resistance to anti-estrogen treatment regimens.

In our clinical experience, the combination of PALBO–FUL in patients with metastatic LBC whose disease relapsed during adjuvant hormonal therapy or progressed after first-line ET for advance disease did meet the expectation. Most patients showed early disease progression and low clinical benefit [15]. We posed the question, why did patients with LBC show lower than reported responses to CDK4/6i? We know that the development and progression of invasive lobular carcinoma (ILC) are characterized by the loss of E-cadherin–E-cadherin binding in normal cells that prevents beta-catenin inhibition of PTEN, thus permitting the inhibition of AKT [19]. As a consequence of the loss of E-cadherin in LBC, the PI3K/AKT pathway is constitutively activated and represents one of the main pathways of proliferation and growth [20,21]. We hypothesized that in patients with metastatic LBC that becomes resistant to endocrine therapy, the hyperactivation of PI3K/AKT signaling may promote an intrinsic resistance to CDK4/6i through the activation of cyclin E/CDK2—amplification of cyclin E is the only factor that showed a correlation with resistance to a CDK 4/6i (palbociclib) in trials [22,23]. Alternatively, inhibition of the AKT pathway could perhaps represent a superior strategy for these patients.

Everolimus is a sirolimus derivative that inhibits mTOR (a key downstream point of the PI3K pathway) through allosteric binding to mTORC1. The results of the BOLERO-2 trial showed that dual-blockade with EVE–EXE more than doubled median PFS versus EXE alone in patients with hormone receptor-positive (HR+)/human epidermal growth receptor 2-negative (HER2−) metastatic BC recurring/progressing on prior non-steroidal aromatase inhibitors (NSAIs) (7.8 versus 3.2 months) [24,25,26]. In addition, results of an Italian observational study suggest that treatment with EXE–EVE is an active and safe therapeutic option for endocrine-sensitive MBC patients in a real-world clinical setting, regardless of treatment lines [27]. These results were confirmed in the BALLET study that enrolled patients more heavily pretreated, with a safety profile consistent with that observed in BOLERO-2 [28]. These results are important because the treatment pattern of MBC is based on the sequence of multiple lines of therapy, and it is therefore vital to determine the possible additive/cumulative effects of different regimens. The combination regimen of EVE and EXE is the only regimen currently registered with an mTOR inhibitor in this setting and represents a valid alternative to the harmful toxicity profile of cytotoxic chemotherapy [29].

In our real-world analysis, median PFS was significantly longer for patients with metastatic LBC receiving EVE–EXE as second-line hormonal treatment compared with PALBO–FUL. Both treatments were well tolerated and only one patient (in the EVE–EXE group) discontinued therapy due to adverse events. Univariate analysis showed that prognosis may be influenced by disease status (de novo metastatic vs. relapsed disease), previous exposure to chemotherapy, and study treatment (PALBO–FUL or EVE–EXE). In particular, patients who had disease relapse and those who received a neo/adjuvant and/or first-line chemotherapy had shorter median PFS, suggesting that de novo metastatic and relapsed disease are characterized by different molecular background which for relapsed cancer is probably the result of clone selection derived from the exposure to previous treatments. The efficacy of chemotherapy in LBC is the subject of much debate, and it is usually reserved for patients with negative prognostic scores, visceral crisis, or when all possible ET lines have been exhausted. Most of our patients received cytotoxic agents (as neoadjuvant/adjuvant), which may have resulted in a detrimental effect, in particular when used in early lines. The PS analysis adjusted for previous chemotherapy exposure and synchronous/metachronous metastatic status confirmed a longer median PFS for patients receiving EVE–EXE (6.0 vs. 4.6 months, *p* = 0.04). Therefore, the activation of the PI3K/AKT pathway in LBC may result in intrinsic resistance to palbociclib after development of refractory disease to prior ET.

The results of this study allow us to speculate that EVE–EXE as a second-line treatment of metastatic LBC may improve therapeutic outcomes. In terms of optimizing sequential therapy, using a CDK4/6i for the first-line treatment of endocrine sensitive tumors is indicated, while mTOR inhibitors could be considered the preferred option when resistance to adjuvant/first-line ET has occurred. Activation of the PI3K/AKT pathway may not solely explain the lack of effectiveness of palbociclib in LBC, and other factors may drive resistance to Palbociclib [30]. Further studies are needed to explore the potential implications of these pathways in the mechanisms of resistance to CDK4/6i. In the era of personalized medicine, improving molecular characterization of cancer to define the best therapeutic program for each patient is paramount.

This retrospective real-world analysis generates the hypothesis of a potential benefit from EVE–EXE in comparison with PALBO–FUL as a second line hormonal-treatment for metastatic luminal breast cancer with lobular histology, and it allows us to speculate on the best therapeutic sequence. However, the limitations of this study, including its retrospective nature and small sample size, need to be addressed.

## 5. Conclusions

The results of this retrospective, real-world analysis seem to suggest a potential benefit of EVE–EXE in comparison with PALBO–FUL as a second-line ET of MBC with lobular histology.

## Figures and Tables

**Figure 1 jpm-10-00291-f001:**
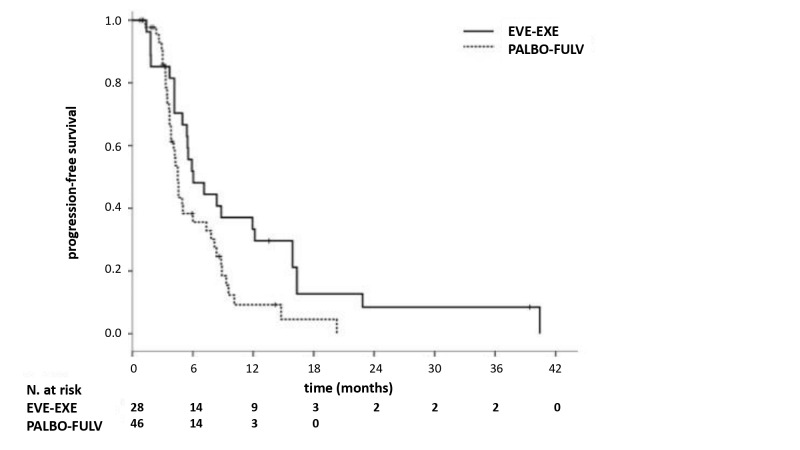
Progression-free survival (PFS). N: number; EVE: everolimus; EXE: exemestane; PALBO: palbociclib; FULV: fulvestrant.

**Figure 2 jpm-10-00291-f002:**
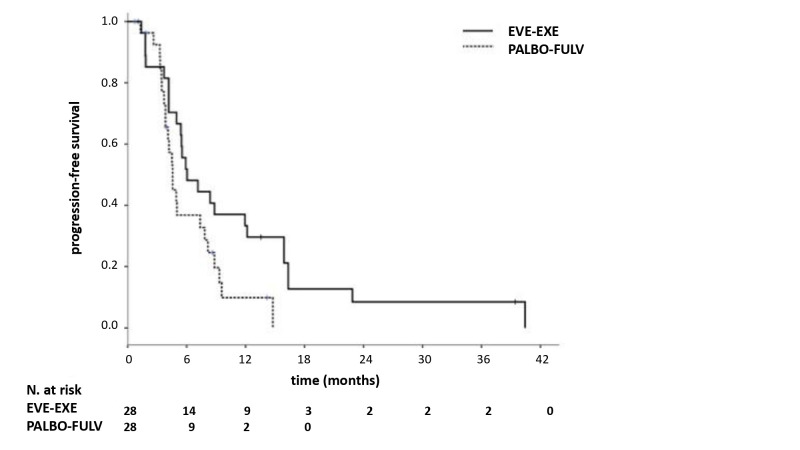
Progression-free survival (PFS) after propensity score adjustment. EVE: everolimus; EXE: exemestane; PALBO: palbociclib; FULV: fulvestrant.

**Figure 3 jpm-10-00291-f003:**
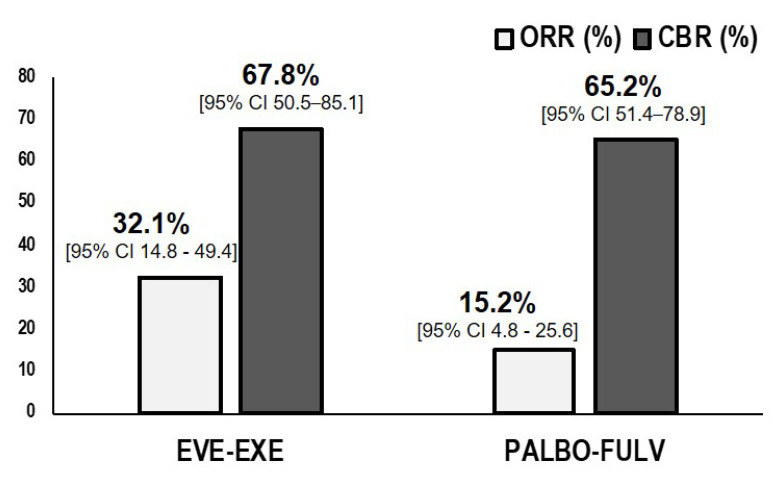
Objective response rate (ORR) according to RECIST 1.1 and clinical benefit rate (CBR). EVE: everolimus; EXE: exemestane; PALBO: palbociclib; FULV: fulvestrant; CI: confidence interval; *p*-value: chi-square.

**Table 1 jpm-10-00291-t001:** Baseline and treatment characteristics (*n* = 74).

Characteristics (*n* = 74)	Palbociclib + Fulvestrant (*n* = 46) (%)	Everolimus + Exemestane (*n* = 28) (%)
Age		
>65	14 (30)	17 (61)
≤65	32 (70)	11 (39)
Menopausal status		
Pre-/Peri-menopausal	5 (11)	0
Post-menopausal	41 (89)	28 (100)
Performance status (ECOG)		
0	23 (50)	12 (43)
1	23 (50)	15 (54)
2	0	1 (3)
Metastatic site		
Bone only	12 (26)	10 (36)
Visceral	18 (39)	9 (32)
Other	16 (35)	9 (32)
Sites of metastasis		
1	19 (41)	15 (54)
2	17 (37)	7 (25)
≥3	10 (22)	6 (21)
Number of previous lines of endocrine therapy		
1	26 (57)	12 (43)
2	20 (43)	16 (57)
Purpose of the most recent treatment		
Adjuvant therapy	13 (28)	1 (3)
Treatment for advanced disease	33 (72)	27 (97)
Disease-free interval		
<12 months	23 (50)	8 (29)
12–24 months	3 (7)	2 (7)
>24 months	13 (28)	9 (32)
Previous endocrine therapies		
Aromatase inhibitors	22 (48)	11 (39)
Tamoxifen	10 (22)	0
Aromatase inhibitors + tamoxifen	14 (30)	7 (25)
Fulvestrant	0	15 (54)
Previous chemotherapy		
Yes	28 (63)	18 (61)
No	18 (37)	10 (39)
Setting of previous chemotherapy		
Neoadjuvant or adjuvant	23 (50)	17 (61)
Advanced disease	5 (13)	1 (3)
Stage at diagnosis		
I	7 (15)	0
II	16 (35)	8 (29)
III	16 (35)	15 (54)
IV	7 (15%)	5 (17)

ECOG: Eastern Cooperative Oncology Group.

**Table 2 jpm-10-00291-t002:** Univariate and multivariate analysis for progression-free survival (PFS).

Characteristics	Univariate	Multivariate
Age (≥65 vs. <65 years)	1.11 (0.66–1.85), *p* = 0.69	-
Menopausal status (post vs. pre)	1.16 (0.42–3.23), *p* = 0.77	-
Metastatic status (synchronous vs. metachronous)	2.82 (1.43–5.56), *p* = 0.003	-
Previous chemotherapy (yes vs. no)	0.41 (0.24–0.72), *p* = 0.002	0.41 (0.24−0.72), *p* = 0.002
Previous hormonal therapy (yes vs. no)	0.67 (0.40–1.13), *p* = 0.13	-
Metastatic sites (visceral vs. not visceral)	1.34 (0.80–2.25), *p* = 0.27	-
Treatment (EVE-EXE vs. Palbo)	0.58 (0.35–0.96), *p* = 0.025	-

**Table 3 jpm-10-00291-t003:** Adverse events for any causes observed during the study period.

Adverse Events from Any Cause	Palbociclib + Fulvestrant (*n* = 46, %)			Everolimus + Exemestane (*n* = 28, %)		
	Any Grade	Grade 3	Grade 4	Any Grade	Grade 3	Grade 4
Any adverse event	35 (76)	22 (48)	2 (4)	24 (85)	6 (21)	0
Neutropenia	30 (65)	18 (39)	2 (4)	3 (10)	0	0
Febrile neutropenia	0	0	0	0	0	0
Anemia	19 (41)	2 (4)	0	4 (14)	0	0
Thrombocytopenia	11(24)	2 (4)	0	2 (7)	0	0
Fatigue	16 (35)	0	0	18 (64)	0	0
Nausea	4 (9)	0	0	5 (18)	0	0
Diarrhea	0	0	0	4 (14)	0	0
Stypsis	4 (9)	0	0	3 (10)	0	0
Headache	1 (2)	0	0	0	0	0
Hot flash	6 (13)	0	0	0	0	0
Stomatitis	1 (3)	0	0	10 (35)	3 (10)	0
Rash	1 (3)	0	0	7 (25)	2 (7)	0
Alopecia	6 (13)	0	0	0	0	0
Myalgia	4 (9)	0	0	2 (7)	0	0
Dyslipidemia and/or Hyperglycemia	0	0	0	4 (14)	0	0
Pneumonitis	0	0	0	6 (21)	1 (4)	0

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
