# Peer review of "Palbociclib Plus Fulvestrant or Everolimus Plus Exemestane for Pretreated Advanced Breast Cancer with Lobular Histotype in ER+/HER2− Patients: A Propensity Score-Matched Analysis of a Multicenter Retrospective Patient Series"

_jpm, 2020, doi:10.3390/jpm10040291_

Round 1

Reviewer 1 Report

The authors provide trial evidence for the efficacy of palbociclib and evorelimus in lobular breast carcinoma.  The study findings could benefit designing treatment for this type of cancer.

How do they distinguish between “previous chemotherapy” and “Previous Chemotherapy Neoadjuvant or Adjuvant Treatment for advanced disease” in table 1? The numbers are different.

Figure 2 please provide legends for the dotted and solid line for KM curve.

Is DFS in table 1 is same as PFS? Why are some patients not included in DFS?

What is ILC in line 200?

Can the authors clarify the line 203, 240 about PI3K/AKT activation and provide reference to the statement?

Why is ORR not significant between groups when it is double in EVo-EXE? If ORR and CBR are similar  what explains the longer PFS in Evo-exe?

Author Response

Pint 1: The authors provide trial evidence for the efficacy of palbociclib and evorelimus in lobular breast carcinoma.  The study findings could benefit designing treatment for this type of cancer.

Response 1: We agree with this statement.

Point 2: How do they distinguish between “previous chemotherapy” and “Previous Chemotherapy Neoadjuvant or Adjuvant Treatment for advanced disease” in table 1? The numbers are different.

Response 2: Thanks for this suggestion. To clarify the two line (10 and 11) of the table 1, we have modified the line 11 ”Previous chemotherapy (new line) Neoadjuvant or Adjuvant (new line) Treatment for advanced disease” in “Setting of previous chemotherapy (new line) Neoadjuvant or Adjuvant (new line) Advanced disease”. We have corrected a typo error (number of patients treated with chemotherapy in Advanced disease in PALBO-FUL arm (5 vs 6) and now the numbers in line 10 and 11 are in agreement.

Point 3: Figure 2 please provide legends for the dotted and solid line for KM curve.

Response 3: Figure 1 has been amended and legends have been added

Point 4: Is DFS in table 1 is same as PFS? Why are some patients not included in DFS?

Response 4: Table 1 shows the DFS ranges of patients enrolled in the study who underwent adjuvant treatment. Because some patients in the study had a disease onset with metastases, the number of patients with DFS reported was less than the total number of patients in the study (39 vs 46 in the PALBO-FUL Arm and 19 vs 20 for EVE- EXE)

Point 5: What is ILC in line 200?

Response 5: thank you for your note, we have updated the text adding Invasive lobular carcinoma (ILC) (line 261)

Point 6: Can the authors clarify the line 203, 240 about PI3K/AKT activation and provide reference to the statement?

Response 6: Thanks for the suggestion, we have added the references below in order to make the discussion more clear and scientifically supported.(line 265)

  1. Lau MT, Klausen C, and Leung, PC. E-cadherin inhibits tumor cell growth by suppressing PI3K/Akt signaling via b-catenin-Egr1-mediated PTEN expression. Oncogene. 2011;30,2753–2766. doi: 10.1038/onc.2011.6.
  2. Liu X, Su L, and Liu X. Loss of CDH1 up-regulates epidermal growth factor receptor via phosphorylation of YBX1 in non-small cell lung cancer cells. FEBS Lett. 2013;587,24, 3995–4000. doi: 10.1016/j.febslet.2013.10.036.

Point 7: Why is ORR not significant between groups when it is double in EVo-EXE? If ORR and CBR are similar  what explains the longer PFS in Evo-exe?

Response 7: ORR appears to be doubled in the EVE-EXE arm compared to PALBO-FUL, however, a statistically significant difference (p 0.07) is not likely to be achieved for the small number of patients. At the same time, CBRs are similar between the two treatment arms for more SD in the PALBO-FUL arm. These events (best ORR for EVE-EXE and CBR similar for SD increase in PALBO-FULV) clearly result in a greater depth of response in the EVE-EXE arm which positively affects the PFS (statistically longer in EVE-EXE).

Reviewer 2 Report

PALBOCICLIB PLUS FULVESTRANT OR EVEROLIMUS PLUS EXEMESTANE FOR PRETREATED ADVANCED BREAST CANCER WITH LOBULAR HISTOTYPE: A PROPENSITY SCORE-MATCHED ANALYSIS OF A MULTICENTER ‘REAL-WORLD’ PATIENT SERIES

This is an observational study comparing two different second line treatment regimens in ER+HER2- lobular breast cancer: palbociclib + fulvestrant vs everolimus + exemestane.

It could potentially help better understand treatment sequences for such patients and inform future clinical trials. However there are several issues:

  1. This is a retrospective observational study. I think the title should be modified to reflect this. ‘real-world’ is not a type of study design. Also ER+HER2- should be included in the title.
  2. There are some typographical (e.g. palociclib) and grammatical errors.
  3. In the introduction section the evidence from randomised trials of the three CDK4/6 inhibitors need to be more thoroughly reviewed. In particular since the focus is on second line treatment, PALOMA3, MONARCH2 and MONALEESA3 are most relevant.
  4. Did these studies report in any subgroup analysis results for lobular histology?
  5. Similarly existing evidence on the efficacy of everolimus should be included in the review.
  6. Lines 76-78: By ‘ER-metastatic’ do you mean metastatic ER+ tumours with endocrine resistance? Please clarify.
  7. Why was palbociclib selected among CDK4/6 inhibitors for this study given that it has shown the least promising results for patients with prior endocrine resistance among CDK4/6 inhibitors?
  8. Lines 108-109: ‘The clinical, biological and pathological characteristics of tumors at baseline were determined using (chi-square test and Fisher's exact test).’ Not sure what this means!
  9. Lines 110-111: ‘Cox proportional hazards model was applied to multivariate survival analysis.’ Same here.
  10. ‘All signifi¬cant variables in the univariate model were used to build the multivariate model of survival. A propensity score (PS) adjustment for baseline characteristics was conducted for survival analysis.’ What are these variables used? They should be listed. Also please provide more details on what is included in the propensity score and how the score was generated. In general the statistical methods should be described in sufficient detail so that if one had the data they would be able to replicate the analysis solely using the information provided in this section.
  11. All estimates from a multivariate model must be reported, not only the one which is statistically significant. The adjusted HR without information on the other variables is meaningless.
  12. The discussion section begins with how tumours eventually become resistant to CDK4/6 inhibitors and lack of evidence by histological subtype, but the corresponding information for everolimus is not given.
  13. Line 204: What does ‘patients who were ER’ mean?
  14. In the discussion/conclusion sections it needs to be made clearer that the findings are only hypothesis-generating given the small sample size, retrospective observational nature of the study and potential selection bias. Also given the lack of detail on how the analysis was done it is not clear whether potential confounding factors have been adequately adjusted for.

Author Response

Point 1: This is a retrospective observational study. I think the title should be modified to reflect this. ‘real-world’ is not a type of study design. Also ER+HER2- should be included in the title.

Response 1: as suggested the title has been amended as follows:

PALBOCICLIB PLUS FULVESTRANT OR EVEROLIMUS PLUS EXEMESTANE FOR PRETREATED ADVANCED BREAST CANCER WITH LOBULAR HISTOTYPE IN ER+HER2- PATIENTS: A PROPENSITY SCORE-MATCHED ANALYSIS OF A MULTICENTER RETROSPECTIVE PATIENT SERIES.

Point 2:There are some typographical (e.g. palociclib) and grammatical errors.

Response 2: thank you for your comment, we have checked and corrected the text accordingly.

Point 3: In the introduction section the evidence from randomised trials of the three CDK4/6 inhibitors need to be more thoroughly reviewed. In particular since the focus is on second line treatment, PALOMA3, MONARCH2 and MONALEESA3 are most relevant.

Response 3: Thanks for your suggestion. We have modified the introduction by focusing on the aspects indicated. (line 81-91)

Point 4: Did these studies report in any subgroup analysis results for lobular histology?

Response 4 : Thanks for your suggestion. We have modified the introduction by focusing on the aspects indicated. (line 81-91)

Point 5: Similarly existing evidence on the efficacy of everolimus should be included in the review.

Response 5: Thanks for your suggestion. We have modified the introduction by focusing on the aspects indicated. (line 81-91)

Point 6: Lines 76-78: By ‘ER-metastatic’ do you mean metastatic ER+ tumours with endocrine resistance? Please clarify.

Response 6: Thank you for your suggestion. We meant ER + metastatic LBC with endocrine resistance, we clarify in the text.(line 92)

Point 7: Why was palbociclib selected among CDK4/6 inhibitors for this study given that it has shown the least promising results for patients with prior endocrine resistance among CDK4/6 inhibitors?

Response 7: We did not specifically select Palbociclib in this analysis, but since Palbociclib was the first CDK4/6i introduced in clinical practice, the data available were all with this molecule.

Point 8: Lines 108-109: ‘The clinical, biological and pathological characteristics of tumors at baseline were determined using (chi-square test and Fisher's exact test).’ Not sure what this means!

Response 8: Thanks for this warning. We use Fisher’s exact test, so we have corrected the typo mistake. (line 142)

Point 9: Lines 110-111: ‘Cox proportional hazards model was applied to multivariate survival analysis.’ Same here.

Response 9: We have added in the text that multivariate survival analysis was done using Cox’s regression model. (line 137)

Point 10: ‘All significant variables in the univariate model were used to build the multivariate model of survival. A propensity score (PS) adjustment for baseline characteristics was conducted for survival analysis.’ What are these variables used? They should be listed. Also please provide more details on what is included in the propensity score and how the score was generated. In general the statistical methods should be described in sufficient detail so that if one had the data they would be able to replicate the analysis solely using the information provided in this section.

Response 10: The PS analysis is conducted to balance the selection bias of a retrospective study, therefore the variables used cannot be postulated at the start of the study, but only on acquired patient data. Therefore, in materials and methods we have described the use of the PS match analysis and in the results we have clearly written the variables used (line 189-190 “PS analysis adjusted for prior chemotherapy and synchronous/metachronous metastatic status”).

Point 11: All estimates from a multivariate model must be reported, not only the one which is statistically significant. The adjusted HR without information on the other variables is meaningless.

Response 11: Thanks for this suggest. However, as represented in many other works in the literature, we considered it clearer to show only the positive value in multivariate, making it easier to read the data.

Point 12: The discussion section begins with how tumours eventually become resistant to CDK4/6 inhibitors and lack of evidence by histological subtype, but the corresponding information for everolimus is not given.

Response 12: Thanks for this suggest. We we added the evidence of efficacy of everolimus in lobular histology in hormone-refractory patients (… and in BOLERO-2 everolimus has been shown to be effective both in ductal and lobular histotype hormone-refractory patients [17]). (line 251)

Point 13: Line 204: What does ‘patients who were ER’ mean?

Response 13: Thanks for this suggest. We meant “patient with metastatic LBC that becomes resistant to endocrine therapy”. We have corrected this mistake (line 264)

Point 14: In the discussion/conclusion sections it needs to be made clearer that the findings are only hypothesis-generating given the small sample size, retrospective observational nature of the study and potential selection bias. Also given the lack of detail on how the analysis was done it is not clear whether potential confounding factors have been adequately adjusted for.

Response 14: Thanks for this suggestion. We have just we have already written in the discussion and conclusion that this work is only hypothesis generator and we believe we have clarified the methods used thanks to your suggestion